# Taming Sparsely Activated Transformer with Stochastic Experts

**Simiao Zuo**[†,*] **, Xiaodong Liu**[◇]**, Jian Jiao**[◇]**, Young Jin Kim**[◇]**, Hany Hassan**[◇]**, Ruofei Zhang**[◇]**,**
**Tuo Zhao**[†] **and Jianfeng Gao**[◇]
[†]Georgia Institute of Technology   [◇]Microsoft
{simiaozuo,tourzhao}@gatech.edu,
{xiaodl,jian.jiao,youki,hanyh,bzhang,jfgao}@microsoft.com

## Abstract

Sparsely activated models (SAMs), such as Mixture-of-Experts (MoE), can easily scale to have outrageously large amounts of parameters without significant increase in computational cost. However, SAMs are reported to be parameter inefficient such that larger models do not always lead to better performance. While most on-going research focuses on improving SAMs models by exploring methods of routing inputs to experts, our analysis reveals that such research might not lead to the solution we expect, i.e., the commonly-used routing methods based on gating mechanisms do not work better than randomly routing inputs to experts. In this paper, we propose a new expert-based model, THOR (**T**ransformer wit**H** St**O**chastic Expe**R**ts). Unlike classic expert-based models, such as the Switch Transformer (Fedus et al., 2021), experts in THOR are randomly activated for each input during training and inference. THOR models are trained using a consistency regularized loss, where experts learn not only from training data but also from other experts as teachers, such that all the experts make consistent predictions. We validate the effectiveness of THOR on machine translation tasks. Results show that THOR models are more parameter efficient in that they significantly outperform the Transformer and MoE models across various settings. For example, in multilingual translation, THOR outperforms the Switch Transformer by 2 BLEU scores, and obtains the same BLEU score as that of a state-of-the-art MoE model (Kim et al., 2021) that is 18 times larger. Our code is publicly available at: https://github.com/microsoft/Stochastic-Mixture-of-Experts.

## 1 Introduction

Large neural network models have shown to be effective in many natural language processing tasks such as machine translation (Lewis et al., 2020; Conneau & Lample, 2019), natural language understanding (Devlin et al., 2019; Liu et al., 2019; He et al., 2020), and natural language generation (Radford et al., 2019; Brown et al., 2020). These models are usually densely activated. That is, a model uses all its parameters to process all inputs. One drawback of these models is the prohibitive training cost. Moreover, the extreme size drastically reduces inference speed, further limiting the models' practicality.

To address these issues, sparsely activated models (SAMs, Shazeer et al. 2017) have been proposed. A SAM adaptively selects a subset of its parameters for different inputs during model training and inference. This makes it possible to train SAMs that are an order of magnitude larger than densely activated models without significant increase in computational cost. For example, the sparsely activated GShard (Lepikhin et al., 2020) consists of over 600 billion parameters and the Switch Transformer (Fedus et al., 2021) 1.5 trillion parameters, while GPT-3 (Brown et al., 2020), which is arguably the largest densely activated model, consists of only 175 billion parameters.

The building block of SAMs is the expert layer, which contains an attention mechanism and multiple feed-forward neural networks (FFNs) in parallel. Each FFN is referred to as an *expert*. During

---

*Work was done during an internship at Microsoft.

training, an input is routed to a fixed number of experts, such that the number of floating point operations (FLOPs) of one forward pass remains constant, regardless of the total number of experts. Thus, training SAMs is much more cost-efficient than training densely activated models. For example, training of Switch-large (Fedus et al., 2021) and that of T5-large (Raffel et al., 2019) require the same forward FLOPs, despite that the former is 35 times larger (26.3 vs. 0.74 billion parameters).

However, SAMs have been reported to be *parameter inefficient*. For example, although the Switch-large model is 35 times larger than T5-large, its performance on the GLUE benchmark (Wang et al., 2019a) is only slightly better (88.5 vs. 87.8). There are also cases where the performance of SAMs is even worse than smaller densely activated models. For example, the performance of Switch-large is worse than T5-large on the ARC Reasoning Challenge (66.0 vs. 68.8) (Clark et al., 2018). In another example, although GShard (Lepikhin et al., 2020) shows substantial gains over densely activated models, a diminishing return with larger number of parameters has been observed.

Most on-going research has focused on improving SAMs by developing effective *routing methods*. Since only a subset of model parameters (i.e., experts) are updated for each input during training, we need to decide which experts to be activated given an input. Existing works (Shazeer et al., 2017; Lepikhin et al., 2020; Fedus et al., 2021; Yang et al., 2021) use a gating network for input routing. However, the gating mechanism suffers from the notorious *load imbalance* issue: the gate's weight could collapse such that nearly all the inputs are routed to the same expert. Therefore, many methods are proposed to mitigate this issue, such as noisy gating (Shazeer et al., 2017), expert capacity (Lepikhin et al., 2020), load balancing loss (Lepikhin et al., 2020; Fedus et al., 2021), and $k$ Top-1 gating (Yang et al., 2021). However, these routing methods have not been proved effective to make SAMs more parameter efficient. To understand why SAMs are not parameter efficient, we analyze the performance of several classic MoE models. Our analysis reveals that a SAM does not always outperform a densely activated model of a similar size, confirming the results reported in Yang et al. (2021). Moreover, we also observe that the widely-used routing method based on the gating mechanism does not work better than randomly routing inputs to experts,

Inspired by our findings, we propose a new SAM, THOR (**T**ransformer wit**H** St**O**chastic Expe**R**ts). Unlike classic SAMs, such as the Switch Transformer, experts in THOR are randomly activated (with no need of any gating mechanism) for each input during training and inference. THOR models are trained by minimizing both the cross-entropy loss and a consistency regularization term, such that experts can learn not only from training data but also from other experts as teachers so that all the experts make consistent predictions.

To validate the effectiveness of THOR, we have conducted extensive experiments on machine translation using three settings: low-resource, rich-resource, and multilingual. Results show that THOR models outperform state-of-the-art MoE models by an average of 2 BLEU score on twelve low-resource translation tasks. In the rich-resource setting, THOR achieves new state-of-the-art results on the two widely-used translation benchmarks, WMT'16 En-De and WMT'14 En-Fr. On multilingual translation tasks, the THOR model with 300 million parameters achieves 2 BLEU score improvement over a state-of-the-art MoE model of the same size. Moreover, our model achieves state-of-the-art results on these tasks — the same BLEU score that is achieved by the Z-code MoE model (Kim et al., 2021) with 5.5 billion parameters (18 times larger).

## 2 BACKGROUND

**Transformer.** The Transformer (Vaswani et al., 2017) model has demonstrated its superior performance in many sequence-to-sequence natural language processing tasks, such as neural machine translation. The model contains an encoder and a decoder. The encoder consists of multiple encoder layers, each having an identical structure. An encoder layer employs a self-attention mechanism and a feed-forward neural network (FFN). The decoder is similarly constructed, except for an additional cross-attention mechanism in each decoder layer.

**Sparsely Activated Models.** The building block of SAMs is the expert layer, which is similar to the Transformer layer. Each of these expert layers contain an attention mechanism and multiple FFNs in parallel, where each FFN is referred to as an *expert*. Let $\{E_i\}_{i=1}^{N}$ denote the experts, and $N$ denotes the total number of experts. A gating mechanism decides to which expert(s) an input should be routed. At each expert layer, given an input vector $x \in \mathbb{R}^d$, where $d$ is the embedding dimension, the gate value of routing $x$ to expert $E_i$ is

$$p_i(x) = [\text{Softmax}(W_g x)]_i, \qquad (1)$$

where $W_g \in \mathbb{R}^{N \times d}$ is the trainable weight matrix of the gating mechanism. Given the gate values $\{p_i(x)\}_{i=1}^N$, we select the top-$K$ experts to form an activated set of experts $\mathcal{T} \subset \{1 \cdots N\}$, where $|\mathcal{T}| = K$. Then the output $x_{\text{out}}$ of the expert layer is

$$x_{\text{out}} = \sum_{i \in \mathcal{T}} p_i(x) E_i(x). \tag{2}$$

Notice that in Eq. 2, input $x$ only activates $K$ instead of $N$ experts, where $K \ll N$, e.g., $K = 2$ and $N = 2048$ in GShard (Lepikhin et al., 2020). This implies that the number of FLOPs required for one forward pass does not increase with the number of experts $N$. Therefore, SAMs can scale to an enormous size without any significant increase in training time and inference time.

The gate weight matrix $W_g$ (Eq. 1) is trained together with the rest of the model parameters. Because there is no constraint on the learned weights, it is possible that $W_g$ collapses such that one row dominates, i.e., all the inputs are routed to one expert. This problem is referred to as *load imbalance*. Existing works adopt various ad-hoc heuristics to mitigate this issue, e.g., adding Gaussian noise to Eq. 1 (noisy gating, Shazeer et al. 2017), limiting the maximum number of inputs that can be routed to an expert (expert capacity, Lepikhin et al. 2020), imposing a load balancing loss (Lepikhin et al., 2020; Fedus et al., 2021), and using linear assignment (Lewis et al., 2021). There are other works that remove the gating mechanism such that load imbalance is no longer an issue, e.g., by incorporating hash functions (Roller et al., 2021). Besides the load imbalance issue, there are also heated discussions on how to construct $\mathcal{T}$ in Eq. 2. For example, Shazeer et al. (2017); Lepikhin et al. (2020); Yang et al. (2021) conjecture that routing inputs to $K > 1$ experts is necessary, while Fedus et al. (2021) argue that using $K = 1$ is sufficient and more computationally efficient.

## 3 ANALYSIS OF SPARSELY ACTIVATED MODELS

We investigate behavior of the gating mechanism of several classic MoE models. We conduct experiments on a multilingual translation task, {De, Vi} $\rightarrow$ En. More details are presented in Appendix A.

We consider two MoE models proposed in Shen et al. (2019), referred to as MoE(dec) and MoE(tok), respectively, and three variants of the Switch Transformer proposed in Fedus et al. (2021). The number of experts is set to two for all the MoE models. We compare them with the Transformer (Vaswani et al., 2017) model of the same model size.

Figure 1 shows the validation losses and BLEU scores of three models: Transformer, MoE(dec), and MoE(tok). We see that the two MoE models perform very similarly, and neither outperforms the Transformer by a significant margin.

To interpret the results of Figure 1, we examine the load of each expert and the confidence scores of routing inputs to different experts. An expert's load is defined as the proportion of inputs that are assigned to it. For an input that is routed to an expert, its routing confidence score (output of the gating mechanism) determines the level of preference, e.g., if the routing confidence score is 0.5, then the gate has no preference for either expert. For each expert, we compute the average routing confidence score over all the inputs assigned to it.

Figure 2 shows that after the early stage of training (i.e., the first 200 iterations), the gate weight collapses and nearly all the inputs are routed to expert 2. Also, the average routing confidence score of expert 2 is close to 1.0, which means that the gate strongly prefers expert 2 to expert 1. In this case, only one of the experts is sufficiently trained. Figure 3 depicts a different scenario, where the inputs are randomly dispatched to the experts. Notice that after approximately 4000 iterations, the two experts are equally loaded, and the probabilities of assigning any input to expert 1 and expert 2 are almost identical, indicating that the gating mechanism has no preference for either expert.

We have identified two behaviors of the gating mechanism: *load imbalance* and *random routing*. The former is also reported in recent papers (Shazeer et al., 2017; Lepikhin et al., 2020; Fedus et al., 2021). We further investigate the Switch Transformer (Fedus et al., 2021), which is a state-of-the-art MoE variant that incorporates various methods to resolve the load imbalance issue. In addition, because behavior of the gating mechanism in the Switch Transformer mimics random routing (see Appendix A), we examine the effect of discarding the gate and randomly assigning inputs to experts. Figure 4 demonstrates the validation losses and BLEU scores of the Transformer and three variants of the Switch Transformer, where inputs are routed according to tokens (referred to as Switch(t)), sentences (Switch(s)), or are routed randomly (Switch(r)). Similar to the results in Figure 1, we

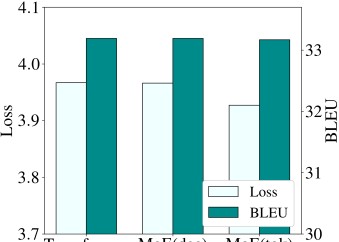

Figure 1: Validation results of MoE(dec) and MoE(tok).

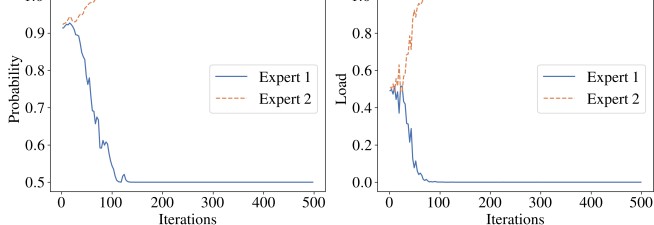

Figure 2: Gating mechanism of MoE(dec). Left: average routing confidence; Right: load of experts.

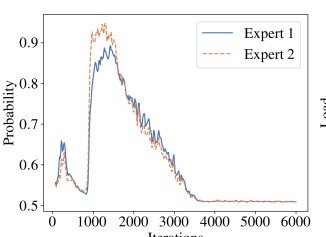
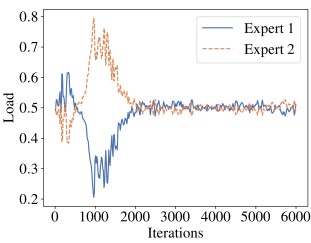
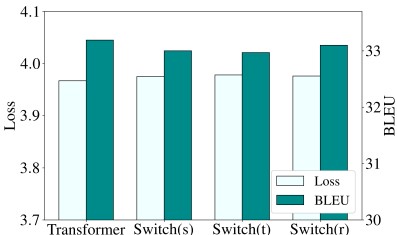

Figure 3: Gating mechanism of MoE(tok). Left: average routing confidence; Right: load of experts.

Figure 4: Performance of three variants of the Switch Transformer.

see that the four models perform similarly. This shows that even after we alleviate load imbalance, model performance is not improved (i.e., the Switch Transformers do not outperform the vanilla Transformer), and the performance of the Switch Transformer does not vary much among different routing methods, including random routing.

We remark that in this paper, we focus on natural language processing tasks, in particular neural machine translation. There are other works in different research fields (e.g., computer vision) that draw different conclusions than ours (Riquelme et al., 2021). We attribute this to the intrinsic differences between image classification and language generation, e.g., each input in the former belongs to a clearly-defined category, while no such knowledge exists in the latter.

In summary, the experiments reveal

- A sparsely activated model does not always outperform a densely activated model of the same model size.
- The widely-used routing method based on the gating mechanism does not work better than randomly routing inputs to experts.

## 4    THOR: TRANSFORMER WITH STOCHASTIC EXPERTS

The ineffectiveness of the gating mechanism, as shown in our experiments, motivates us to propose a new expert-based model, THOR (**T**ransformer wit**H** St**O**chastic Expe**R**ts). In THOR, a pair of experts are randomly selected and activated in each layer during a training iteration, and then all the inputs in a batch are processed using the same pair of experts. Our method drastically simplifies model design, and has two additional advantages. First, it eliminates the load imbalance issue because randomly selecting a pair of experts in each iteration allows each expert to have a fair chance to be sufficiently trained. The ad-hoc heuristics, such as the load balancing loss, as discussed in Section 2, are no longer needed. Second, unlike the gating mechanism, THOR does not introduce any additional model parameters.

One problem of THOR is that without a gating mechanism, experts need to be randomly selected during inference, and we may obtain inconsistent inference results due to different random seeds. For example, on a Czech-to-English translation dataset, our experiments show that randomness can result in a 0.5 BLEU score difference.

To address this issue, we introduce a consistency regularizer in the training objective of THOR. Concretely, let $N$ denotes the number of experts, $L$ the number of layers, and $E_i^l$ an activated expert

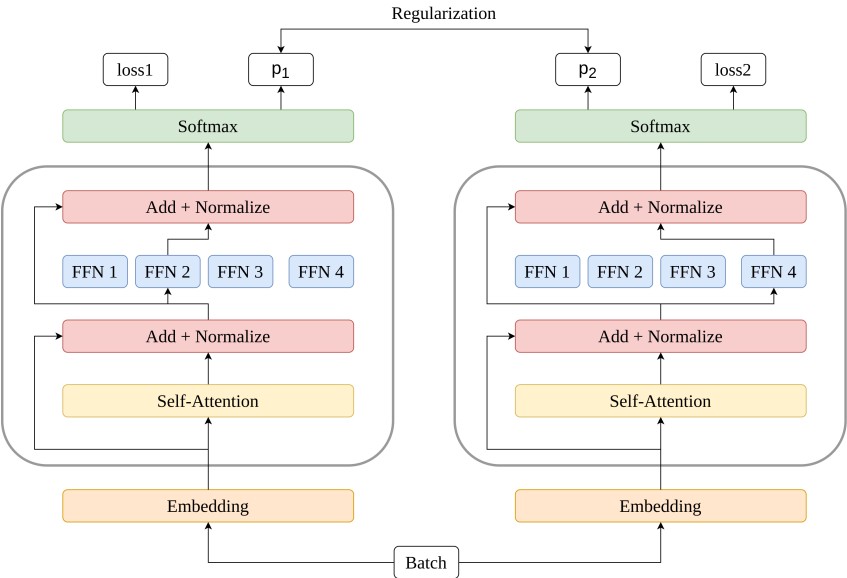

Figure 5: Illustration of a training iteration with stochastic experts. For conciseness, we show a model with only one Transformer layer.

(which is a FFN) in layer $l$, where $1 \leq i \leq N$ and $1 \leq l \leq L$. We use $p = f(x; \{E_i^l\}_{l=1}^L)$ to indicate the prediction probability of input $x$ using the model $f$ where experts $\{E_i^l\}_{l=1}^L$ are activated. Figure 5 illustrates one training iteration. Notice that instead of activating one expert for each layer in an iteration, we select to activate a pair of experts in THOR. As a result, we obtain two prediction probabilities produced by the two selections, respectively: $p_1 = f(x; \{E_i^l\}_{l=1}^L))$ and $p_2 = f(x; \{E_j^l\}_{l=1}^L))$. Then, the training objective of THOR with respect to training samples $(x, y)$ in the dataset $\mathcal{D}$ is

$$\min \sum_{(x,y)\in\mathcal{D}} \ell(x,y) = \mathrm{CE}(p_1; y) + \mathrm{CE}(p_2; y) + \alpha\mathrm{CR}(p_1; p_2),$$

$$\text{where } \mathrm{CR}(p_1; p_2) = \frac{1}{2}\left(\mathrm{KL}(p_1\|p_2) + \mathrm{KL}(p_2\|p_1)\right).$$

(3)

Here, CE is the cross-entropy loss, the consistency regularizer CR is defined as the average of the two Kullback–Leibler (KL) divergence terms, and $\alpha$ is a hyper-parameter that controls the strength of the regularizer. In mini-batch SGD training, we randomly sample a pair of experts to activate at each layer for each batch. During inference, we can also randomly select an expert to activate at each layer for each input, similar to that in training. We can also use different expert-selection methods, such as expert-ensemble, as to be discussed in Section 5 (Table 5).

The THOR training objective of Eq. 3 forces all the experts to minimize training errors while making the same predictions as much as possible. Thus, in each training step, each expert optimizes its parameters by learning from both the training data (via minimizing the cross-entropy loss) and its paired expert as a teacher (via minimizing the KL divergence). Although these experts are learned to make consistent predictions, they converge to different (local) optima given the randomness introduced in training, e.g., initialization, mini-batch SGD, random routing, etc. Thus, every expert learns from a set of diverse teachers during the course of training, which helps to improve model's performance. In addition, by penalizing experts that yield inconsistent predictions from the others, the consistency regularizer also helps reducing the variance of model prediction.

THOR is conceptually similar to dropout (Srivastava et al., 2014) since both methods route an input to some randomly selected sub-net components (i.e., experts in THOR and neurons in dropout). However, THOR differs from dropout in several important aspects, making it a better choice for efficient training and serving of large-scale neural models. First, THOR can be applied to both training and inference, while dropout is only used for training. Second, THOR is shown to be more robust in large-scale model training than dropout. For example, our models are less likely to overfit with the increase in the number of experts (see Figure 9). Third, THOR leads to a sparse model

that is more structured than that of dropout, such that a large-scale THOR model can be much more easily trained using GPU clusters, e.g., by putting different experts on different GPUs in parallel.

## 5 EXPERIMENTS

We evaluate THOR on neural machine translation. We adopt three settings: low-resource translation, rich-resource translation, and multilingual translation. For low-resource and rich-resource translation, we train all the models using *Fairseq*[1] (Ott et al., 2019). For multilingual translation, we use *DeepSpeed MoE*[2] (Kim et al., 2021) to implement the MoE models. All the experiments are conducted on NVIDIA V100 GPUs. Additional experiments, including model scale-up and comparison of inference speed, are deferred to Appendix D.

### 5.1 BASELINE

We use two baselines in the experiments.

- Transformer (Vaswani et al., 2017) achieves superior performance in many sequence-to-sequence learning tasks, such as neural machine translation.
- Switch Transformer (Fedus et al., 2021) is a state-of-the-art MoE model, which employs a gating mechanism to route inputs and uses a load balancing loss to reduce load imbalance.

To verify the effectiveness of the imposed consistency regularizer in Eq. 3, we also compare THOR with Transformer models trained using two popular regularization methods. We remark that these two methods share similar computational costs with THOR, i.e., they also require two forward passes in each training iteration.

- SMART (Jiang et al., 2020) utilizes a smoothness inducing adversarial regularizer to penalize the worst case difference between predictions of a clean input and a perturbed input.
- R3F (Aghajanyan et al., 2020) uses a regularizer to reduce representational collapse. The method has shown to be effective in various natural language processing tasks.

All the methods are trained for the same number of FLOPs in the experiments for fair comparison.

### 5.2 LOW-RESOURCE TRANSLATION

We use six language pairs: English to Vietnamese, English to German, and English to French from IWSLT; English to Romanian, English to Latvian, and English to Czech from Europarl[3]. Dataset statistics are summarized in Table 6 (Appendix B).

Table 1: Experimental results on low resource datasets. The best result on each dataset is in **bold**.

|                                       | En-Vi | Vi-En | En-De | De-En | En-Fr | Fr-En |
|---------------------------------------|-------|-------|-------|-------|-------|-------|
| Transformer (Vaswani et al., 2017)    | 31.3  | 29.4  | 28.1  | 34.8  | 39.2  | 38.1  |
| SMART (Jiang et al., 2020)            | 32.5  | 30.5  | 29.3  | 35.8  | 40.0  | 38.8  |
| R3F (Aghajanyan et al., 2020)         | 32.2  | 30.7  | 29.2  | 35.7  | 39.7  | 38.9  |
| Switch (Fedus et al., 2021)           | 31.7  | 29.5  | 28.4  | 34.6  | 39.1  | 38.2  |
| THOR                                  | **34.0** | **33.0** | **31.1** | **37.8** | **40.7** | **40.0** |
|                                       | En-Ro | Ro-En | En-Lv | Lv-En | En-Cs | Cs-En |
| Transformer (Vaswani et al., 2017)    | 23.5  | 25.0  | 13.6  | 15.8  | 16.1  | 20.4  |
| SMART (Jiang et al., 2020)            | 24.6  | 25.7  | 14.2  | 16.3  | 16.7  | 21.4  |
| R3F (Aghajanyan et al., 2020)         | 23.8  | 25.8  | 14.4  | 16.3  | 16.8  | 21.6  |
| Switch (Fedus et al., 2021)           | 23.8  | 24.4  | 13.8  | 16.1  | 16.1  | 20.6  |
| THOR                                  | **25.2** | **27.1** | **15.2** | **17.4** | **17.6** | **22.4** |

---

[1] https://github.com/pytorch/fairseq
[2] https://github.com/microsoft/DeepSpeed
[3] https://www.statmt.org/europarl

To evaluate THOR with different model sizes, we use the Transformer-base (Vaswani et al., 2017) architecture on Europarl datasets, and a smaller model on IWSLT datasets. Compared with Transformer-base, the smaller model decreases the hidden dimension from 2048 to 1024, and decreases the number of heads from 8 to 4 with the dimension of each head doubled. We use two experts for the expert-based models. We remark that even though THOR increases the number of parameters, its inference speed (in terms of FLOPs) is the same as Transformer-base because only one expert is activated for each input. Interested readers refer to Appendix C for more details.

The experimental results in Table 1 show that performance of the Switch Transformer is on par with the vanilla Transformer, e.g., its average BLEU score on the 12 datasets is 26.3, the same as the Transformer. The results confirm that SAMs do not outperform densely activated models with similar model sizes. In contrast, THOR achieves more than 1.0 BLEU score improvement over the Switch Transformer in all the 12 tasks. THOR also significantly outperforms the models trained using the two competing regularization methods, SMART and R3F.

## 5.3 RICH-RESOURCE TRANSLATION

We use two widely adopted rich-resource translation benchmarks: English to German translation from WMT'16 and English to French translation from WMT'14. The former dataset consists of 4.5 million training sentence pairs, and the latter 36 million pairs. We follow the pre-processing steps in Ott et al. (2018).

To evaluate THOR , We use the Transformer-big architecture (Vaswani et al., 2017) and we set the number of experts for both THOR and the Switch Transformer to 4. Interested readers refer to Appendix C for more details.

Table 2 reports the BLEU scores and the sacre-BLEU scores (Post, 2018) of different models. We see that THOR achieves new state-of-the-art results in the setting where neither data augmentation nor pre-trained language model is used. Specifically, THOR lifts the previous state-of-the-art (Liu et al., 2020b;c) by 0.3 BLEU score on the En-De translation task and 0.1 BLEU score on the En-Fr translation task. THOR also significantly outperforms the models trained using the other two regularization methods, SMART (Jiang et al., 2020) and R3F (Aghajanyan et al., 2020). Similar to what is observed in low-resource translation, the Switch Transformer (Fedus et al., 2021) does not outperform the vanilla Transformer (Ott et al., 2018).

Table 2: BLEU and sacreBLEU scores on WMT'14 En-Fr and WMT'16 En-De. Results of Jiang et al. (2020), Aghajanyan et al. (2020), and Fedus et al. (2021) are from our implementation.

| BLEU | En-De | En-Fr |
|---|---|---|
| Vaswani et al. (2017) | 28.4 | 41.8 |
| Ott et al. (2018) | 29.3 | 43.2 |
| Wang et al. (2019b) | 29.6 | — |
| Wu et al. (2019a) | 29.7 | 43.2 |
| So et al. (2019) | 29.8 | 41.3 |
| Jiang et al. (2020) | 29.8 | 43.4 |
| Wu et al. (2019b) | 29.9 | 43.3 |
| Aghajanyan et al. (2020) | 29.4 | 43.3 |
| Liu et al. (2020c) | 30.1 | **43.8** |
| Fedus et al. (2021) | 29.3 | 43.0 |
| THOR | **30.4** | **43.8** |

| sacreBLEU | En-De | En-Fr |
|---|---|---|
| Ott et al. (2018) | 28.6 | 41.4 |
| Jiang et al. (2020) | 29.1 | 41.5 |
| So et al. (2019) | 29.2 | — |
| Aghajanyan et al. (2020) | 29.0 | 41.5 |
| Liu et al. (2020c) | 29.5 | 41.8 |
| Fedus et al. (2021) | 28.6 | 41.1 |
| THOR | **29.6** | **41.9** |

## 5.4 MULTILINGUAL TRANSLATION

We have collected 10 language pairs from WMT datasets, and built a $64k$-entry dictionary for all the languages. The detailed statistics are summarized in Table 7 (Appendix B). Please refer to Kim et al. (2021) for more details. We do not use multi-task learning or additional monolingual data in the experiments.

We use the following model architecture: the embedding dimension is set to 768 and the hidden dimension for the FFN is set to 3072; we use 12 encoder layers and 6 decoder layers, where each layer has 12 attention heads, and the dimension of each head is 64. We set the number of experts to 4 for both THOR and the Switch Transformer.

Table 3 reports the average BLEU score of translating English to other languages, translating other languages to English, and the overall score of the 20 tasks. We see that compared with the Switch

Transformer of the same size (i.e., 300 million parameters), our model achieves a 2-point improvement in the overall BLEU score. In addition, our model is far more parameter efficient than the Switch Transformer. The THOR model with 300 million parameters achieves the same BLEU score (24.4) that is achieved by the Switch Transformer with 5.5 billion parameters, which is more than 18 times larger.

Table 3: Multilingual translation results. Here "$E$" means the number of experts.

|  | En→Others | Others→En | Average |
|---|---|---|---|
| Switch (32E, 5.5B) | — | — | **24.4** |
| Switch (4E, 300M) | 20.3 | 24.6 | 22.4 |
| THOR (4E, 300M) | **21.4** | **27.4** | **24.4** |

Figure 6 shows BLEU scores in all the 20 translation tasks. Notice that THOR outperforms the baseline on 17 out of the 20 tasks. The improvement is in general more significant on the tasks with smaller datasets. For example, our model achieves BLEU score improvement of 4.7 and 6.7 on Gu-En ($85k$) and Hi-En ($264k$), respectively. On the tasks with larger datasets, the improvement obtained by our model is less substantial, but still significant, e.g., $+0.9$ BLEU score on Cs-En ($10M$) and $+1.1$ Fi-En ($4.8M$). For the only three tasks where our model underperforms the baseline, the gaps are small, e.g., $-0.4$, $-0.2$, and $-0.4$ BLEU scores on En-Cs, En-De, and En-Fr, respectively.

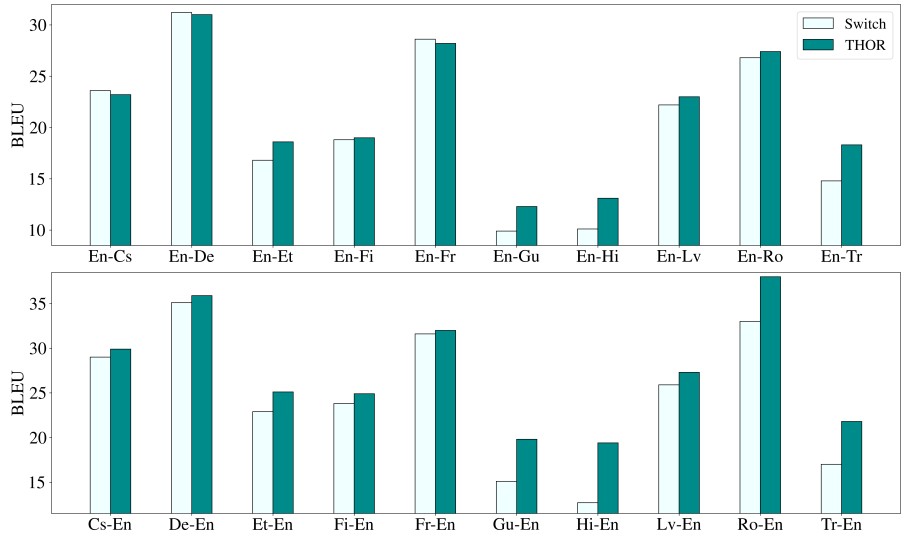

Figure 6: Details of multilingual translation results.

## 5.5 ABLATION EXPERIMENTS

**Training Objective.** We examine the relative contributions of the three loss terms used in the THOR training objective of Eq. 3: $CE_1$, $CE_2$ and $CR$. The result in Table 4 shows that the consistency regularizer $CR$ is crucial to the model performance, and that dropping one of the two CE terms leads to only very small BLEU score loss since the two cross-entropy terms play the same role in training.

**Inference Methods.** We compare three inference methods: (1) `Dispatch(s)` uses sentence-level random routing, where all tokens in one sentence are routed to the same expert; (2) `Dispatch(t)` uses token-level random routing, where tokens within a sentence are routed to different experts; (3) `Ensemble`, where each sentence is routed to all the $N$ experts, and the $N$ hidden representations in each layer are averaged. Note that the number of FLOPs is larger for `Ensemble` because we need to run forward pass for each input through $N$ experts. Table 5 shows that `Dispatch(s)` and `Dispatch(t)` perform similarly, and `Ensemble` yields the best BLEU score with a cost of longer inference time.

Table 4: Effect of the three loss terms in training object of Eq. 3, tested on Cs-En translation.

| Loss terms | BLEU |
|---|---|
| $CE_1 + CE_2 + CR$ | 22.4 |
| $CE_1 + CR$ | 22.2 |
| $CE_1 + CE_2$ | 20.8 |
| $CE_1$ | 20.6 |

Table 5: Performance and costs of three inference methods, tested on Cs-En translation.

| | BLEU | time |
|---|---|---|
| Dispatch(s) | 22.4 | $\times 1$ |
| Dispatch(t) | 22.4 | $\times 1$ |
| Ensemble | 22.6 | $\times N$ |

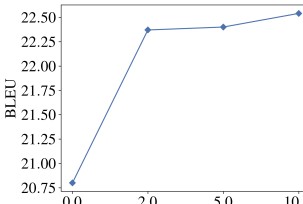

Figure 7: Effect of the consistency regularization strength $\alpha$ on Cs-En translation.

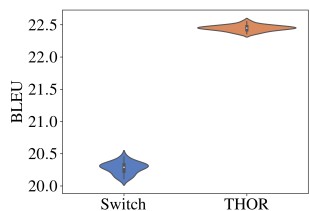

Figure 8: Violin plot of performance consistency on Cs-En translation.

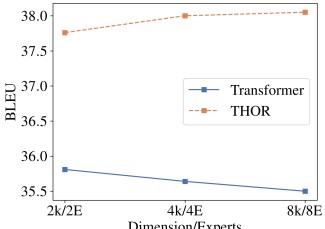

Figure 9: BLEU vs. model size on De-En translation.

**Regularization strength.** To investigate the effect of the regularization strength $\alpha$, we run experiments on the Cs-En translation dataset in the low-resource setting. Figure 7 shows that model performance is not very sensitive to $\alpha$ as long as the value is large enough, say $\alpha > 2.0$.

**Consistency of Model Prediction.** We study the variance of model prediction due to the use of randomly activated experts during inference. We compare THOR and the Switch Transformer, where we remove the trained gate during inference. For each model, we compute the variance of model prediction based on 20 runs. As shown in Figure 8, THOR makes more consistent predictions than Switch Transformer due to the use of the consistency regularizer for model training. The variance of THOR is below $0.002$, whereas the variance of Switch Transformer is $0.008$, four times larger. We remark that by removing the trained router from the Switch Transformer, model performance only marginally decreases (from $20.6$ to $20.4$). This further indicates that a trained router may not be better than a random router.

**Overfitting.** We compare the THOR model and the Transformer model regarding how likely they overfit the training data when the model size increases. We run experiments on the De-En data in the low-resource setting, where the dropout rate of the FFNs in the Transformer is selected such that the number of parameters trained in one iteration is the same as the THOR model. As shown in Figure 9, THOR does not show any sign of overfitting — we observe a consistent improvement in BLEU score as we increase the number of experts from 2 to 8. In contrast, the Transformer model's performance deteriorates as we increase the hidden dimension of its FFN from $2k$ to $8k$. We remark that we also observe the overfitting phenomenon on larger datasets, e.g., the Transformer overfits on the Cs-En dataset when we set the hidden dimension of its FFN to $16k$.

## 6 CONCLUSION

We present a new expert-based sparsely activated model, THOR. Unlike existing SAMs, such as the Switch Transformer, experts in THOR are randomly activated for each input during training and inference. THOR models are trained using a consistency regularized loss, where every expert learns not only from training data but also from other experts as teachers so that all the experts make consistent predictions. As a result, not only can large-scale THOR models be trained and served as efficiently as classic MoE models, THOR models also demonstrate a better generalization capability in that they are more parameter-efficient, less likely to overfit, make more consistent predictions, and achieve better results consistently across different settings. We validate the effectiveness of THOR via a comprehensive empirical study on machine translation. In all the three settings (i.e., low-resource, rich-resource, and multilingual translation), THOR models significantly outperform the vanilla Transformer, and Switch Transformer, a state-of-the-art MoE model.

## ACKNOWLEDGMENTS

We thank Rukmini Lyer, Kevin Duh, Hao Cheng, Chunyuan Li, Johannes Gehrke, colleagues from Microsoft Bing Ads team and Microsoft Research for their valuable discussions and comments.

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

# A    ANALYSIS OF SPARSELY ACTIVATED MODELS

## A.1    TRAINING DETAILS

We consider two Mixture-of-Experts (MoE) models proposed in Shen et al. (2019), which are denoted "MoE(dec)" and "MoE(tok)". In the first variant, each expert is a separate Transformer decoder. In the second variant, each expert is a different token, i.e., if we route the input to expert one, then we replace the $\langle bos \rangle$ (begin-of-sentence) token in the input sentence with a $\langle expert_1 \rangle$ token. Note that embeddings of these expert tokens are trained together with the rest of the model parameters. These models are equipped with an expectation-maximization optimization framework. Such a framework facilitates computing the probability of assigning an input to a specific expert according to the gating mechanism. Please refer to Shen et al. (2019) for details about these models.

We use a multilingual translation setting, where we adopt two datasets: De-En from IWSLT'14 and Vi-En from IWSLT'15. For each dataset, we use byte pair encoding (BPE, Sennrich et al. 2016) with $10,000$ merge operations for pre-processing. Then we concatenate the two pre-processed datasets. We learn a separate dictionary for En and {De+Vi}, which resulted in approximately $9k$ and $12k$ vocabularies, respectively.

For training, we use Adam (Kingma & Ba, 2015) as the optimizer and we set the learning rate to $0.001$. We set the batch size to be equivalent to $64k$ tokens, e.g., we use $8k$ tokens per GPU with $8$ GPUs. Other training details follow the *Fairseq*[4] implementation. For inference, we use a beam size of 5 and a length penalty of $1.0$.

## A.2    ADDITIONAL RESULTS

We also plot the average routing confidence score and the load of experts for Switch(s) and Switch(t), similar to Figure 2 and Figure 3. We first investigate the Switch Transformer without the load balancing loss.

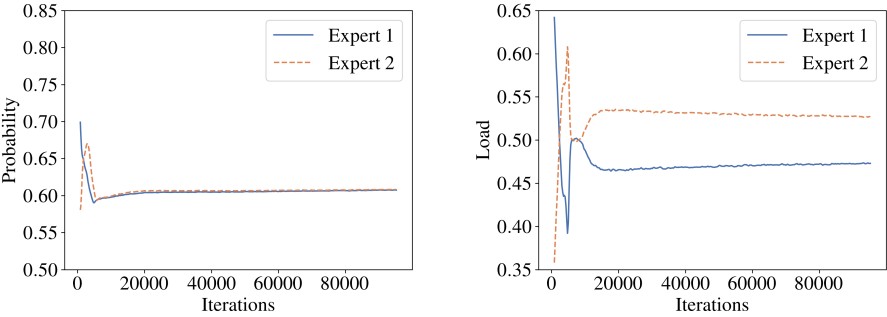

Figure 10: Switch(s) w/o load balancing. Left: average routing confidence; Right: load of experts.

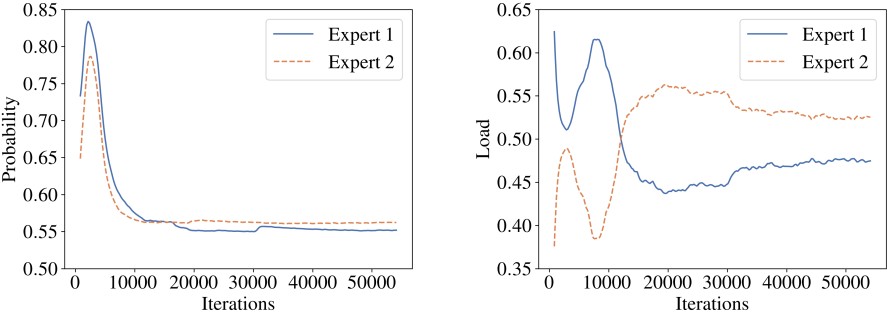

Figure 11: Switch(t) w/o load balancing. Left: average routing confidence; Right: load of experts.

---

[4]https://github.com/pytorch/fairseq/blob/master/examples/translation/

Figure 10 shows the results for Switch(s) without the load balancing loss, where we route inputs to experts on the sentence-level. We see that after about $10k$ training iterations, the average routing confidence score of expert 1 and expert 2 becomes similar, and both of these scores are around $0.60$. Moreover, the load of the experts are not balanced, i.e., there is a $10\%$ difference in the loads ($55\%$ vs. $45\%$). We conclude that behavior of the gating mechanism of Switch(s) is similar to Figure 3, i.e., the gate is essentially randomly routing inputs to experts without any preference.

Figure 11 shows the results for Switch(t) without the load balancing loss, where we route inputs to experts on the token-level, i.e., different tokens within the same sentence may be routed to different experts. Similar to the Switch(s) case, the average routing confidence score of both of the two experts converges to around $0.55$. This indicates that the gate do not prefer any expert given an input. Moreover, the load of the experts are not balanced, the same as in Figure 10. Based on these observations, we conclude that behavior of the gating mechanism of Switch(t) is also *random routing*.

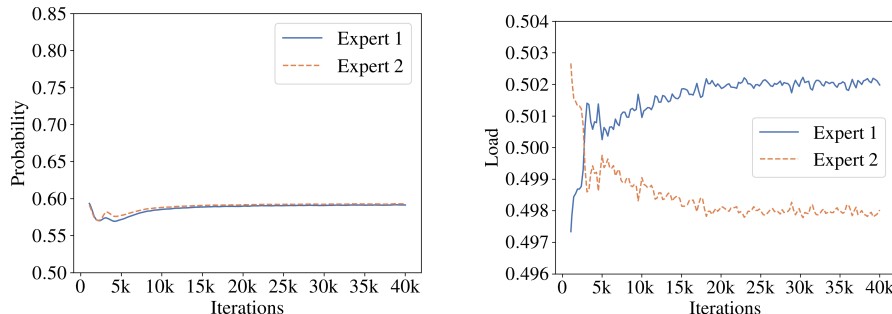

Figure 12: Switch(s) w/ load balancing. Left: average routing confidence; Right: load of experts.

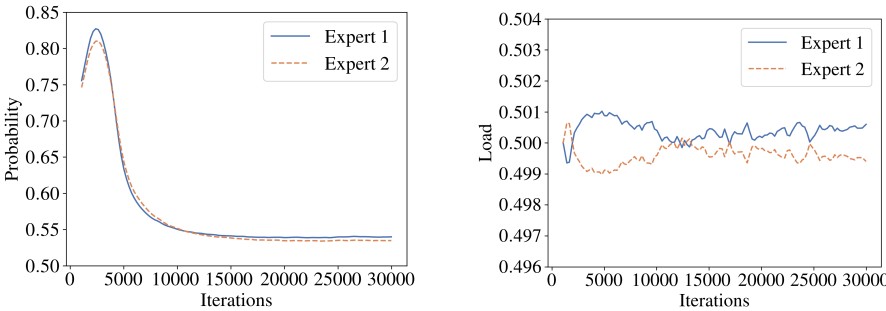

Figure 13: Switch(t) w/ load balancing. Left: average routing confidence; Right: load of experts.

Figure 12 and Figure 13 show behavior of the gating mechanism of Switch(s) and Switch(t) equipped with the load balancing loss, respectively. We see that the load balancing loss indeed balances the load for both Switch(s) and Switch(t), e.g., there is a less than $0.4\%$ imbalance for Switch(s) and less than $0.2\%$ imbalance for Switch(t). In comparison, the imbalance is around $10\%$ for the two Switch Transformer variants without the load balancing loss. Also, similar to the case without the load balancing loss, the average routing confidence score converges to around $0.60$ for Switch(s) and around $0.55$ for Switch(t). Based on the observations, we conclude that behavior of the gating mechanism is still *random routing* when Switch(s) and Switch(t) are equipped with the load balancing loss.

## B    DATASETS

Statistics of low-resource datasets are shown in Table 6. The English-Vietnamese, English-German, and English-French datasets are from[5] IWSLT'14, '15, and '16, respectively. The training data of

---

[5]https://iwslt.org/

English-Romanian, English-Latvian, and English-Czech are from Europarl[6], and the validation and testing data are from WMT'17.

Statistics and data sources used in the multilingual translation task are shown in Table 7.

Table 6: Statistics of low resource translation datasets.

|  | En-Vi | En-De | En-Fr | En-Ro | En-Lv | En-Cs |
|---|---|---|---|---|---|---|
| Train | 117,055 | 160,239 | 218,256 | 390,746 | 591,631 | 619,029 |
| Validation | 5,098 | 7,283 | 8,453 | 1,900 | 1,949 | 2,902 |
| Test | 1,268 | 6,750 | 1,133 | 1,999 | 2,001 | 3,005 |

Table 7: Statistics of multilingual translation datasets. The other language in the translation tasks is English (En) for all the datasets.

| Language | Czech (Cs) | German (De) | Estonian (Et) | Finnish (Fi) | French (Fr) |
|---|---|---|---|---|---|
| Data source | WMT'19 | WMT'19 | WMT'18 | WMT'19 | WMT'15 |
| # Samples | 10,273,696 | 4,613,192 | 695,227 | 4,838,576 | 9,999,995 |

| Language | Gujarati (Gu) | Hindi (Hi) | Latvian (Lv) | Romanian (Ro) | Turkish (Tr) |
|---|---|---|---|---|---|
| Data source | WMT'19 | WMT'14 | WMT'17 | WMT'16 | WMT'18 |
| # Samples | 85,688 | 264,199 | 1,444,235 | 540,562 | 182,269 |

## C  TRAINING DETAILS

### C.1  LOW RESOURCE TRANSLATION

We build a joined dictionary for the source and target languages for each dataset. To facilitate this, we use byte pair encoding (BPE) with $10,000$ and $40,000$ split operations for the IWSLT and the WMT datasets, respectively. Other pre-processing steps follow the *Fairseq* implementation.

For training, the regularization strength is chosen to be $\alpha = 5.0$. We set the batch size to be equivalent to $32k$ tokens, i.e., if we have four GPUs, then we set the number of tokens on each GPU to be $4k$ and accumulate gradients for two steps. We use Adam as the optimizer with $\beta_1 = 0.9$, $\beta_2 = 0.98$, and we set the learning rate to be $0.0015$. We train the model for $40k$ steps, and we test the model that yield the highest validation BLEU. For validation and testing, we use a beam size 5 and a length penalty $1.0$. Other training and inference details follow the *Fairseq* implementation.

### C.2  RICH RESOURCE TRANSLATION

Strength of the consistency regularizer is set as $\alpha = 2.0$. We use Adam (Kingma & Ba, 2015) as the optimizer with $\beta_1 = 0.9$, $\beta_2 = 0.98$, and the learning rate is chosen as $0.001$. For inference, we use a beam size 4 and a length penalty $0.6$ for En-De; we use a beam size 10 and a length penalty $1.0$ for En-Fr. Other post-processing steps follow Ott et al. (2018). We report both the BLEU score and the sacreBLEU score (Post, 2018), where the latter is a safer token-agnostic version of BLEU.

### C.3  MULTILINGUAL TRANSLATION

For training, we set the batch size to be equivalent to 1.6 million tokens, e.g., 4096 tokens per GPU with 24 GPUs, and we accumulate gradients for 16 steps. We use RAdam (Liu et al., 2020a) as the optimizer with parameters $\beta_1 = 0.9$ and $\beta_2 = 0.98$. The learning rate is set to be $0.05$. Also, we set the dropout ratio to be $0.1$, and we use label smoothed cross entropy (Szegedy et al., 2016) with a smoothing factor $0.1$. The regularization strength is set to be $\alpha = 4.0$. For inference, we use a beam size 5 and a length penalty $1.0$.

---

[6]https://www.statmt.org/europarl/

# D ADDITIONAL EXPERIMENTS

We further test behavior of THOR and the Switch Transformer when we increase the number of experts. To avoid overfitting, we use a small model (*Transformer-IWSLT*) on the WMT'16 En-De translation dataset. In this experiment, the Transformer model has $48M$ parameters, models with 2, 16, and 64 experts have $55M$, $143M$, and $456M$ parameters, respectively.

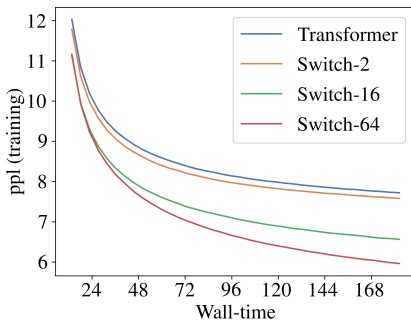 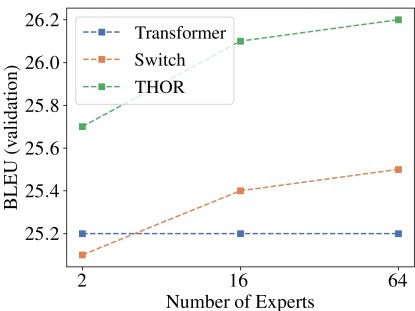

Figure 14: Effects of the number of experts on WMT'16 En-De translation. Left: training perplexity (lower the better) with respect to wall-time (measured in GPU hours); Right: validation BLEU (higher the better) after training for 180 GPU hours with respect to the number of experts, where the size of *Transformer* does not change.

Figure 14 demonstrates the results. In Figure 14 (left), notice that the Switch Transformer trains faster than the vanilla Transformer, and this scaling property is more significant when we increase the number of experts.

From Figure 14 (right), we see that with 2 experts, the Switch Transformer behaves slightly worse the vanilla Transformer in terms of validation BLEU. However, when we increase the number of experts, performance of the Switch Transformer continues to improve and outperforms the vanilla Transformer with the same number of FLOPs. This indicates that in order for a sparsely activated model to outperform a densely activated one, we need to scale the former to contain much more parameters than the latter. Our observations are consistent with existing literature (Lepikhin et al., 2020; Fedus et al., 2021). For example, in Fedus et al. 2021, the sparsely activated Switch-base outperforms the densely activated T5-base using the same number of FLOPs. However, the former is more than 30 times larger (7.5 billion vs. 0.22 billion parameters).

Our method is more *parameter efficient* than the conventional methods. From Figure 14 (right), we see that THOR significantly outperforms the vanilla Transformer and the Switch Transformer even with only 2 experts. Moreover, when we increase the number of experts, performance of THOR also improves.

We also compare inference speed of Transformer, Switch Transformer, and THOR in Table 8. Note that for THOR , we use the `Dispatch(s)` method in Table 5. Note that the inference speed of Switch Transformer and THOR is slower than the vanilla Transformer because of the computation and communication overhead induced by input routing. Such an overhead is more noticeable when the number of experts is large. We remark that in Fedus et al. 2021, the speed of Switch-base is about half of T5-base (780 vs. 1600 samples per second).

Table 8: Inference speed (tokens/second).

| # experts | Transformer — | Switch 2 | 16 | 64 | THOR 2 | 16 | 64 |
|---|---|---|---|---|---|---|---|
| Speed | 15.2k | 15.0k | 10.4k | 7.4k | 15.1k | 10.6k | 7.5k |

