# OpenReview forum: "Taming Sparsely Activated Transformer with Stochastic Experts"
_ICLR.cc/2022/Conference — ICLR 2022 Poster_

### Official Review · Reviewer_PXVZ · 2021-10-22

**Correctness:** 3
**Technical Novelty And Significance:** 3
**Empirical Novelty And Significance:** 4
**Recommendation:** 6
**Confidence:** 5

**Main Review:**

The paper is very well written and clear, proposing the THoR method, including ablations on aspects of its design, and performing three classes of evaluations, comparing to other methods, reporting that THoR outperforms all other methods and establishes at least two state-of-the-art results on WMT. Fundamentally, I am more than willing to believe the key results which show that THoR outperforms Switch Transformer.

However, MoEs are complex models, for which it is very easy to subtly apply an incorrect comparison, or to evaluate improperly. There are several results in the paper which claim that Switch Transformer underperforms or just equals the performance of a dense, standard transformer. I find this difficult to reconcile with the results of the Switch Transformer paper [1], which not only claims that Switch outperforms dense models, but has several results (Figure 1A, and Appendix D) specifically saying increasing the number of experts from 1 to larger numbers monotonically improves performance. A general result which has been mirrored in other recent MoE-like papers (Figure 3 in [2], Figure 6 in [3], Figure 4 in [4]). This inconsistency makes me skeptical of the hierarchy of methods suggested by this paper.

Fundamentally, the main request of this rebuttal is that the authors address this inconsistency between their MoE/Switch results and those reported by others. I have three concrete asks:

* Are methods FLOP-matched when being compared, or step-matched? My read of sections 4 and 5 are that methods are step-matched, but this seems substantially unfair since THoR (and the other regularization methods) require twice the compute per step. I believe a proper comparison across all the datasets and evaluators should be at a fixed number of FLOPs (i.e., with Switch/Dense running for double the number of steps). In line with this request, the inclusion of validation curves showing the performance of THoR, Switch and Dense Transformers as training progresses would be very valuable.

* Are the authors able to reproduce Figure 1A from [1]? An exact reproduction isn't necessary, differences in model details or dataset are fine, but it would be extremely valuable to know whether or not the author's implementation of Switch mirrors the result that increasing the number of experts should increase the performance of the network. If yes, what is your explanation for Switch not outperforming a Dense transformer on some evaluations? If no, do you have an explanation for the difference? Does THoR exhibit improvement with more experts?

*  This paper uses just 2 or 4 experts for the majority of its experiments. While Switch does already claims performance improvements with so few experts, it is also much smaller than the normal regime which is tested, with number of experts in the 16-128 regime being standard [1, 2, 3, 4, 5]. What do the initial experiments, as well as the later Switch vs THoR evaluation comparisons, look like with a larger number of experts? Does the comparison hold up?

If the authors address these three points and either explain or reconcile the difference in their reported Switch performance with those of other papers, then I would have no problem recommending an accept (8). In the current form, I feel this paper is borderline due to these unanswered questions.




Additionally, I have several smaller nits and comments:

* **[General]** There are several recent methods which claim to improve over switch ([2] and [5] come to mind). I don't necessarily think you need to compare against them, but as the claimed performance improvement is high, it seems important to acknowledge.

* **[General]** The authors prefer the term SAM, but this is a very general term (there are lots of different sparsely-activated models out there) and this term doesn't seem standard in the literature. Maybe something more specific would be more apt?

* **[Section 2, "Transformer"]** "the model" should be replaced with "our model", not all transformers are encoder/decoder models.

* **[Section 2, "Sparsely Activated Models"]** A nit: Hash Layers don't help solve the load balance issue, they completely obviate the need for a routing layer (also, from Section 4, worth noting that similar to THoR, Hash layers also don't need to introduce new parameters).

* **[Appendix A.2]** I think the conclusions you're reaching are too certain. While it is true that random routing is a possibility based on these results, it is not a foregone conclusion (what manifests as a slight preference in the router might be key). A conclusive experiment you could run would be to replace the routing decisions *only at evaluation time* by random decisions, and see how much the evaluation performance of the networks are impacted.

* **[Section 4]** The idea that THoR proposes, that experts should be similar and consistency-regularized towards each other, has a very reasonable interpretation. However, it is directly in opposition to the standard interpretation (that experts should be diverse). I think it would be good to discuss this difference more.

* **[Section 4, "different experts on different GPUs in parallel"]** Can you elaborate on this? I understand expert parallelism, but a key aspect of expert parallelism applied to most MoE models is the expectation that for any training minibatch, we expect to be using all experts more or less equally (which means we can share the compute equally across a bunch of different devices, each with their own experts). In THoR, for any given step we're fixing the number of experts we use to two, so I don't see how this can be effectively parallelized across more than 2 devices (without replication).

* **[Section 5.4, Table 3]** Having numbers from a dense transformer would be useful here.

* **[Section 5.5, Figure 8]** Switch should have input jitter disabled at evaluation time (Figure 15 in [1]). This should make the network at eval deterministic. What is the explanation for the variance seen in this figure?


[1] *Switch Transformer* https://arxiv.org/abs/2101.03961

[2] *Hash layers* https://arxiv.org/abs/2106.04426

[3] *GShard* https://arxiv.org/abs/2006.16668

[4] https://arxiv.org/abs/2109.10465

[5] *BASE Layers* https://arxiv.org/abs/2103.16716

**Summary Of The Paper:**

The paper proposes THoR, an approach towards training MoE-like models (called SAMs in the paper) that have multiple internal experts which are chosen in a discrete fashion. Unlike other common methods, which generally incorporate a form of router which maps input to an  specific router choice, THoR makes random, global, per-minibatch decisions on experts, and includes a cross-expert regularization term to help align experts to each other. This paper reports that THoR outperforms other methods, including the recent Switch Transformer MoE-style model, on several classes of multilingual translation tasks.

**Summary Of The Review:**

The paper is well written and the proposed ideal seems novel. But my inclination to recommend accept (8) is tainted by some doubts about the primary comparison given with Switch Transformer.  I've outlined three key questions/additions which could assuage these doubts.

---

> ### Author Response · Authors · 2021-11-16
> **Thank you for the feedback. We have revised the experiment setting and added new experiments. (Part 1 of 2)**
>
> **General comments**
>
> Our claim that MoE (e.g., the Switch Transformers) does not outperform dense models of the **same size** does not contradict claims in other literature. For example, in the Switch Transformer paper, all the comparisons are FLOP-matched instead of parameter-matched. In Figure 1 of the Switch Transformer paper, T5-base has 223M parameters, while Switch-base has 7.4B parameters (see Table 5 in their paper). Our premise suggests that if we have a dense model that also has 7.4B parameters, it will have equally good or even better performance than Switch-base.
>
> **New experiments**
>
> 1. The methods are step-matched. We remark that the two regularization methods (SMART and R3F) that we compare share the same FLOPs as THOR, therefore comparisons with these methods are fair. For Transformer and Switch, we obtain new results by training these models for the same FLOPs as THOR.
> The new results are updated in Table 1 (colored red), and the remaining results will be updated once we finish the experiments. Note that model performance only marginally improves with the longer training time. For example, for the Vi-En experiment, performance of Switch increases by 0.1 BLEU while that of Transformer remains the same.
> 2. Yes, we are able to reproduce the results from the Switch Transformer paper. We have added several experiments in Appendix D. Specifically, we train Switch models with 2, 16 and 64 experts, respectively. Indeed, Switch trains faster than vanilla Transformer. Moreover, by adding more experts Switch outperforms the vanilla Transformer. This is consistent with existing literature. We highlight that [1], even though Switch-base outperforms T5-base, it has over 30 times more parameters, i.e., Switch-base uses 16 experts and has 7.5B parameters, while T5 has 223M parameters. In our experiments (Table 1), the MoE model with 2 experts behaves on par or slightly worse than the vanilla Transformer. However, by adding more experts Switch outperforms its dense counterpart.
> 3. We also conducted experiments on THOR using 2, 16 and 64 experts (Appendix D). Similar to the results in Table 1, our method significantly outperforms Switch. Moreover, performance of THOR improves as we add more experts, similar to Figure 9. The validation BLEU is summarized in the below table.
> |  | Transformer | Switch-2 | Switch-16 | Switch-64 | THOR-2 | THOR-16 | THOR-64 |
> |---|:---:|:---:|:---:|:---:|:---:|:---:|:---:|
> | BLEU | 25.2 | 25.1 | 25.4 | 25.5 | 25.7 | 26.1 | 26.2 |

---

> ### Author Response · Authors · 2021-11-16
> **Thank you for the feedback. We have revised the experiment setting and added new experiments. (Part 2 of 2)**
>
> **Other comments**
>
> 1. (General) The purpose of Base and Hash layers is to improve **training efficiency**, while our method targets **parameter efficiency**, i.e., how to make better use of the model parameters. Therefore, Base and Hash cannot substantially improve model performance given a fixed model size, although they will improve training efficiency.
>
> 2. (General) The term *Sparsely-Activated Models* is from the Switch Transformer paper. We avoided using the term Mixture-of-Experts (MoE) because our method (and the Switch Transformer) do not contain the *mixture* component, e.g., only one expert is used for each input during inference.
>
> 3. (Section 2) In this paper, because we consider sequence-to-sequence generation tasks, we refer to the encoder-decoder model in (Vaswani et al. 2017) when we talk about *Transformer*.
>
> 4. (Section 2) Thank you for pointing this out. We have revised the paper.
>
> 5. (Appendix A.2) We have conducted the suggested experiment in Figure 8, where we remove the trained router during evaluation for Switch. From Figure 8, we see that validation performance only marginally decreases (from 20.6 to 20.4), which further supports our claim that the trained router may not be better than a random router.
>
> 6. (Section 4) In THOR, we still want the experts to learn different knowledge (i.e., different hidden representations). The purpose of the consistency regularizer is to encourage different experts to make consistent predictions. That is, we want each expert to learn knowledge on its own, and we also want all the experts to serve the same purpose. On the other hand, the trained experts also exhibit variability. The consistency regularizer only partially eliminates the variance, i.e., from Figure 8, THOR still shows variability. Moreover, because of different initialization, experts converge to different final parameters, indicating that they indeed learn differently.
>
> 7. (Section 4) THOR shares the same computational advantage as conventional MoE models. For example, suppose we have a model with 128 experts, which is too large to fit in a single GPU. Expert parallelism works by putting one set of experts on different GPUs, instead of letting each GPU to host an entire set of experts, e.g., GPU0 contains expert [0,1] and GPU1 contains expert [2,3], instead of each GPU contains [0,1,2,3]. In a training iteration, we send the batch to 2 experts, e.g., expert0 on GPU0 and expert3 on GPU1. The same procedure is implemented for other MoE models.
>
> 8. (Section 5.4) Due to the limited time provided during the rebuttal period, we will provide these results later. The dataset used is very large and training a dense model requires some tuning efforts. However, based on our experiments of a similar model size (Table 2), we believe performance of the dense Transformer will be similar to the Switch Transformer.
>
> 9. (Section 5.5) In Figure 8, for the Switch Transformer, we use a router during model training and remove it during inference. In this way, we can inspect whether the experts truly have “expertise”. For example, if by removing the router model performance drastically decreases, then the router is useful. From Figure 8, we see that validation performance only marginally decreases (from 20.6 to 20.4), which further supports our claim that the trained router may not be better than a random router.

---

> ### Author Response · Authors · 2021-11-17
> **More experiments on Switch with random routing.**
>
> We also examine performance of random routing with more experts. We use the same setting as in Appendix D (Figure 14), where we set the number of experts to 16. We train a Switch model without router, i.e., use random routing, and we use 20 different random seeds during inference. We summarize the statistics in the below table. Note that the Switch Transformer with a trained router also has a 25.4 BLEU.
>
> |  | Mean | Median | Min | Max | Std |
> |---|:---:|:---:|:---:|:---:|:---:|
> | Switch-16 (random) | 25.4 | 25.4 | 25.3 | 25.6 | 0.1 |

---

### Official Review · Reviewer_hGSm · 2021-10-31

**Correctness:** 4
**Technical Novelty And Significance:** 4
**Empirical Novelty And Significance:** 4
**Recommendation:** 8
**Confidence:** 4

**Details Of Ethics Concerns:**

I don't see any ethics issues.

**Main Review:**

The paper is easy to follow and the proposed methods are simple and novel. The experiments and ablation study are extensively conducted. The results look promising and the method should be easy to be adopted.

I have a few questions and comments.

1. In Table 2, Transformer-base (Vaswani et al.,2017) architecture is used for Europarl datasets, while a smaller model is used on IWSLT datasets. I am wondering why not apply both model architecture to both datasets.

2. In Table 2, it seems to me that the size of Switch transformer is larger than transformer-base but they have same inference flops? However, I also see this statement `The results confirm that SAMs do not outperform densely activated models with similar model sizes.` in Sec. 5.2. So I am wondering if the model size of switch transformer is same as the size of transformer-based model in Table 2.

3. In page 5, the paper states `During inference, we can also select a pair of experts to activate at each layer for each input, similar to that in training.`. I thought only one expert is activated during inference to gain the inference speed. Also, if two experts are selected, how the final predictions are selected? Is the average embedding used?

4. The paper mostly compares the inference FLOPS. I am wondering if the authors can provide the real inference time for transformer-base model and the THOR model (with same FLOPS) in both batch inference mode and real time inference mode.

**Summary Of The Paper:**

This paper proposed to use randomly selected experts for Mixture-of-Experts models instead of gating function based selection methods. To avoid large performance variance of the random selection during inference, the paper proposed to add a consistency regularization which drives the similarity between different experts. The experimental results show its superior performance over existing MoE models such as switch transformers.

**Summary Of The Review:**

The paper presented a simple and novel MoE method. The experiments and ablation study are extensively conducted. The results look quite promising and the method should be easy to be adopted. I vote for acceptance.

---

> ### Author Response · Authors · 2021-11-16
> **Thank you for the feedback. New experiments are added.**
>
> Thank you for the comments.
> 1. The settings are standard in neural machine translation literature, where a small model is used for IWSLT, e.g., [1,2]. \
> [1] https://arxiv.org/abs/1901.10430 \
> [2] https://arxiv.org/abs/1905.11901
> 2. In Table 2, the Switch Transformer has more parameters than the vanilla Transformer. This is because each layer in the Switch Transformer has 4 experts. However, only one of the experts is used for each input, therefore the Switch Transformer has similar inference FLOPs with the vanilla Transformer. It is customary to compare models with the same FLOPs, e.g., in the Switch Transformer paper, Switch-base (7.4B parameters) is compared with T5-base (223M parameters).
> 3. This is a typo, we mean “randomly select one expert to activate at each layer…”. It is possible to use multiple experts. For example, in Table 5, the ensemble method is where we use all the experts during inference. In this case, the output of the experts are averaged in each layer.
> 4. We have added the results in Appendix D. The results are also shown in the below table. Note that the inference speed of Switch Transformer and THOR is slower than the vanilla Transformer because of the computation and communication overhead induced by input routing. Such an overhead is more noticeable when the number of experts is large. We remark that in the Switch Transformer paper, the speed of Switch-base is about half of T5-base ($780$ vs. $1600$ samples per second).
> |  | Transformer | Switch-2 | Switch-16 | Switch-64 | THOR-2 | THOR-16 | THOR-64 |
> |---|:---:|:---:|:---:|:---:|:---:|:---:|:---:|
> | Speed | 15.2k | 15.0k | 10.4k | 7.4k | 15.1k | 10.6k | 7.5k |
>
> **More experimental results**
>
> We have added several new experiments in Appendix D. Specifically, we train Switch and THOR models with 2, 16 and 64 experts, respectively. Similar to the results in Table 1, our method significantly outperforms Switch. Moreover, performance of THOR improves as we add more experts, similar to Figure 9. The validation BLEU is summarized in the below table.
>
> |  | Transformer | Switch-2 | Switch-16 | Switch-64 | THOR-2 | THOR-16 | THOR-64 |
> |---|:---:|:---:|:---:|:---:|:---:|:---:|:---:|
> | BLEU | 25.2 | 25.1 | 25.4 | 25.5 | 25.7 | 26.1 | 26.2 |

---

> ### Comment · Reviewer_hGSm · 2021-11-29
> **I've read all reviews and authors' responses. I'll keep my score.**
>
> All my concerns except for No. 4 are addressed by authors' response. The added inference speed table partially addresses my concern No. 4. I'd suggest to add more details about the setting for Table 8, e.g. number of GPUs, batch size, etc.
>
> One common concern that other reviews have is about the opposite claim against the traditional understanding about sparse models, i.e.,  experts are learning similarly with each other. The experimental results look reasonable to me. So I think this paper provides a new understanding and discovery about sparse models, which may help future research.

---

### Official Review · Reviewer_zyWV · 2021-11-02

**Correctness:** 3
**Technical Novelty And Significance:** 2
**Empirical Novelty And Significance:** 2
**Recommendation:** 5
**Confidence:** 4

**Main Review:**

This paper studies sparse models in the context of deep learning: neural networks that only activate a subset of all parameters depending on the input. This is a field that is gaining lots of attention recently, as models have grown large, expensive and inference time has increased accordingly.

So far, most works [1, 2, 3] learn a router (usually a linear map + softmax + maybe noise) to assign tokens to experts. This approach has some downsides (lack of differentiability, expert collapse that requires additional auxiliary losses, etc). The paper proposes a new approach that simply samples the experts at random --and enforces some coherence among them.

Section 3 presents some experiments related to standard gating mechanisms, Section 4 presents the algorithm, and Section 5 shows the experimental results.

I have concerns in the three sections.

**Section 3.**

The number of experts is 2 --and I assume the model selects 1--, which seems maybe too few to me, in order to see generalizable effects.

If I understand Figures 1-3 correctly, two MoE models were trained (MoE(dec) and MoE(tok)). The first one collapses (only Expert 2 ends up being used) in Figure 2, and the second one seems to reach an equilibrium, in Figure 3. I think the interpretation provided by the paper for Figure 2 is generally wrong and misleading. The fact that both experts are used equally often (i.e. load ~ 0.5 in both cases), and the fact that the *average* routing confidence is 0.5 does *not* imply random routing is happening, as suggested in the paper. If uniformly random routing was happening, we'd see this. However, there are good routing schemes that would lead to this averaged outcome too. To give a simple example, suppose we have a dataset containing sentences about sports and about animals, half and half. The router may send all sports tokens to Expert 1 with weight 1.0, and all animal tokens to Expert 2 with weight 1.0. The average gating weight is 0.5 for each expert, and they get half the tokens. The routing, however, is very far from random.

I'd not agree with the general statement (and, more importantly, I don't think this paper contains enough evidence to back it up): "The widely-used routing method based on the gating mechanism does not work better than randomly routing inputs to experts".

There are published experiments (see [3], Figure 27, layer 21) that replace a trained router with a random router and, as expected, performance goes down. Of course, this does not imply that if routing is random while training, final performance will be worse (but we'll end up with an ensemble of similar experts). Moreover, there's evidence that, under learned routing, experts specialize (even in a human understandable way), see [3] Figure 7. These facts should probably be acknowledged in the paper. Also, note that even fixed routing [4] is fundamentally different from random routing, it still allows for expert specialization if done well.

**Section 4.**

The proposed algorithm seems to weaken the sparse models advantage, that is, having lots of capacity which the model can specialize per input, while keeping cost kind of constant. By randomly selecting experts in every batch (+ explicitly adding a consistency loss), we are forcing all experts to learn the same thing --and how to deal with all possible inputs/tokens--, which directly counteracts the sparse models core idea. In other words, we are training a computationally-cheap ensemble. This can still be a very good model, nonetheless; but I'd be surprised if it beats a properly tuned MoE transformer (more on this in the next section). As mentioned before, there's evidence that experts specialize [3].

Also, I was wondering if there's concrete evidence to support this claim in page 5: "Although these experts are learned to make consistent predictions, they converge to different (local) optima given the randomness introduced in training [...]". I'd assume there's such type of statements in the ensemble literature.

**Section 5.**

I have some fundamental concerns with the experimental section. If my understanding of the experiments is correct, the comparison between THOR and the Switch Transformer [2] is not fair. Basically, the number of epochs is fixed (or say, datapoints seen by the models during training, batch size * number of steps). However, THOR applies two times the backbone network to each datapoint, leading to a massive mismatch in terms of total FLOPs or runtime (with equivalent hardware).

In order to have a meaningful comparison, all the tables in the paper should include two additional columns: total training FLOPs and total runtime. This will unhide compute differences across models.

Switch uses k=1 (i.e. one expert selected per token). Note that even for k=2 (as used in [1, 3]) previous sparse models would be cheaper (and significantly so) than THOR, as all the common --non-experts-- layers are only applied once, regardless of k. One baseline that must be in the paper tables is a Switch model that runs for a FLOPs- or runtime-matched amount of steps, that is, it probably runs for around twice as many steps. I'm open to change my score/reviews if those experiments are added, and are still favorable to THOR.


Another couple of comments regarding experiments. First, for non-ensemble methods to shine (i.e. Switch), probably more than 2/4 experts would help. Using so few experts probably helps mitigate the lack of specialization effect, and benefits the ensemble-like THOR approach.

Second, I don't understand the following sentence at the end of section 5.3: "Similar to what is observed in low-resource translation, the Switch Transformer does not outperform the vanilla Transformer". Looking at Table 2, it seems BLEU for Switch is (quite?) better than for Vaswani et al (i.e. "vanilla" Transformer)?



[1] - GShard: Scaling Giant Models with Conditional Computation and Automatic Sharding

[2] - Switch Transformers: Scaling to Trillion Parameter Models with Simple and Efficient Sparsity

[3] - Scaling Vision with Sparse Mixture of Experts

[4] - Hash Layers For Large Sparse Models

**Summary Of The Paper:**

The paper proposes a new routing mechanism for sparse models in the context of language tasks. Rather than learning a parametric router that learns how to assign tokens to experts, the proposed algorithm (THOR), randomly selects two experts per mini batch, and applies those experts independently to every input. A consistency loss is used to force experts to provide similar predictions. A number of experiments are provided suggesting the algorithm outperforms previous works.

**Summary Of The Review:**

Experiments devote significantly more compute to THOR than to Switch. Accordingly, it is hard to interpret the results.

------

I've raised my score given the authors rebuttal and updated results (3 --> 5).

---

> ### Author Response · Authors · 2021-11-16
> **Thank you for the feedback. We have revised the experiment setting and added new experiments.**
>
> **Section 3**
>
> We believe there are some mis-understanding about the reported metric. As stated in our paper (fourth paragraph in our paper), “for each expert, we compute the average routing confidence score over all the inputs assigned to it”. For example, if tokens about sports are routed to Expert 1 with confidence 1.0, and tokens about animals are routed to Expert 2 with confidence 1.0, then the average confidence of each expert is 1.0 instead of 0.5. This is because we only compute confidence of tokens that are actually routed to a specific expert.
>
> Our claim that random routing will not drastically hurt performance is supported by evidence. In Figure 8 (left violin), we replace the trained router in the Switch-Transformer by a random router. As demonstrated, performance only marginally decreases (from 20.6 to 20.4).
>
> We appreciate the reviewer for pointing out this paper [3]. However, the vision Transformer is considering a fundamentally different task than ours. Training of the vision Transformer is a classification task, whereas we consider a generation task (e.g., machine translation). That is, in paper [3], each input belongs to a clearly-defined class, which is not the case in generation tasks. In our setting, the definition of human interpretable knowledge is very vague, e.g., we cannot claim a specific token (or a sentence) belongs to any category. Therefore, because we consider a fundamentally different setting than the vision Transformer, the conclusions drawn in [3] are not transferable to our setting, and vice versa.
>
> There are many efforts trying to transfer algorithms/experience from the NLP community to the CV community, and vice versa. However, this does not mean conclusions drawn from the same method are always the same. We believe the observed different behavior of the router in MoE will motivate subsequent research. We have highlighted that our paper focuses on NLP applications such as translation, and we have also added discussions about the suggested paper [3].
>
> **Section 4**
>
> The premise of our method is that the router may not behave better than random routing. This is supported by our analysis in Section 3. There is no clear evidence (at least in natural language processing) that the router can automatically learn useful knowledge (refer to our response above). Therefore, the proposed method does not weaken the advantages of sparse models.
>
> Because of different parameter initialization, experts indeed converge to different final parameters. By manually inspecting a trained model, the norm of the weight matrix in different experts exhibits large variance. Also, from the view of ensemble, the consistency regularizer only partially eliminates the variance, i.e., from Figure 8, THOR still shows variability. This further indicates that different experts converge to different parameters.
>
> **Section 5**
>
> 1. **FLOP-matched comparison.** We remark that the two regularization methods (SMART and R3F) that we compare share the same FLOPs as THOR, therefore comparisons with these methods are fair. For Transformer and Switch, we obtain new results by training these models for the same FLOPs as THOR.
> The new results are updated in Table 1 (colored red), and the remaining results will be updated once we finish the experiments. Note that model performance only marginally improves with the longer training time. For example, for the Vi-En experiment, performance of Switch increases by 0.1 BLEU while that of Transformer remains the same.
>
> 2. **More experts.** We have added several new experiments in Appendix D. Specifically, we train Switch and THOR models with 2, 16 and 64 experts, respectively. Similar to the results in Table 1, our method significantly outperforms Switch. Moreover, performance of THOR improves as we add more experts, similar to Figure 9. The validation BLEU is summarized in the below table.
> |  | Transformer | Switch-2 | Switch-16 | Switch-64 | THOR-2 | THOR-16 | THOR-64 |
> |---|:---:|:---:|:---:|:---:|:---:|:---:|:---:|
> | BLEU | 25.2 | 25.1 | 25.4 | 25.5 | 25.7 | 26.1 | 26.2 |
>
> 3. **Transformer baseline.** The results of (Vaswani et al. 2017) are copied from the original Transformer paper. In our paper, we use the Fairseq [1] implementation of Transformer (Ott et al. 2018), which is more efficient. Therefore, we use the results of (Ott et al. 2018) as the baseline for the vanilla Transformer. \
> [1] https://github.com/pytorch/fairseq/blob/main/examples/scaling_nmt/README.md

---

> > ### Comment · Reviewer_zyWV · 2021-11-30
> > **Thank you for the response.**
> >
> > I'd like to thanks the authors for the detailed response, and the updated results, which I find convincing.
> >
> > I've raised my score (3 --> 5) and I won't oppose paper's acceptance.

---

> ### Author Response · Authors · 2021-11-17
> **More experiments on Switch with random routing.**
>
> We also examine performance of random routing with more experts. We use the same setting as in Appendix D (Figure 14), where we set the number of experts to 16. We train a Switch model without router, i.e., use random routing, and we use 20 different random seeds during inference. We summarize the statistics in the below table. Note that the Switch Transformer with a trained router also has a 25.4 BLEU.
>
> |  | Mean | Median | Min | Max | Std |
> |---|:---:|:---:|:---:|:---:|:---:|
> | Switch-16 (random) | 25.4 | 25.4 | 25.3 | 25.6 | 0.1 |

---

### Official Review · Reviewer_PQMm · 2021-11-02

**Correctness:** 3
**Technical Novelty And Significance:** 3
**Empirical Novelty And Significance:** 3
**Recommendation:** 6
**Confidence:** 3

**Main Review:**

Strength
1 The paper is well written with good structure
2 The method is concise and novel to some degree.

Weakness
1 The number of experts is limited to 2. Indeed it would maintain the computation load. But the power of experts may not be exhibited sufficiently. It would be great to investigate the performance by changing the number of experts.
2 I am wondering if this mechanism can be used in the vision transformers?

**Summary Of The Paper:**

In this paper, the authors propose to equip transformer with stochastic experts ( i.e., a number of FFN layers in parallel)  to boost model capacity without increasing much computation. Moreover, they propose a consistency regularization between a pair of experts in the training, which can alleviate performance deterioration of random expert selection in the inference.


**Summary Of The Review:**

Basically, it is an OK paper for me. The novelty and experiments are good to show effectiveness.

---

> ### Author Response · Authors · 2021-11-16
> **Thank you for the feedback. New experiments are added.**
>
> **Weakness 1:**
> 1. Our method also works for a larger number of experts. As a demonstration, in Figure 9, we show that model performance continues to increase when we increase the number of experts.
>
> 2. We have also added several new experiments in Appendix D. Specifically, we train Switch and THOR models with 2, 16 and 64 experts, respectively. Similar to the results in Table 1, our method significantly outperforms Switch. Moreover, performance of THOR improves as we add more experts, similar to Figure 9. The validation BLEU is summarized in the below table.
>
> |  | Transformer | Switch-2 | Switch-16 | Switch-64 | THOR-2 | THOR-16 | THOR-64 |
> |---|:---:|:---:|:---:|:---:|:---:|:---:|:---:|
> | BLEU | 25.2 | 25.1 | 25.4 | 25.5 | 25.7 | 26.1 | 26.2 |
>
> **Weakness 2:**
>
> Our method works for the Transformer model in general. Therefore, we believe the proposed algorithm should work for training vision Transformers.

---

### Public Comment · ~Ye_Bai1 · 2022-02-09
**Three small questions**

Thank you for your nice work! The paper is very well written, and the motivation is very strong.
I have three small questions.
1. To compute consistency loss, the network needs two-pass forward propagation. Does it need much more time to train the whole network?
2. The experts is selected randomly during inference. Can we say all the experts are important equally? If they are equally important, can we remove the last N-1 experts and only use the first expert for inference? Because it can save spaces of model.
3. I found that the consistency loss brings essential gain of the MoE. Recently, RDrop also uses KLD to make randomly sampled models consistent. Did you compare your method with R-Drop?


Ref. Wu, Lijun, et al. "R-drop: regularized dropout for neural networks." Advances in Neural Information Processing Systems 34 (2021).

---

### Decision · Program_Chairs · 2022-01-20

**Decision:**

Accept (Poster)

**Comment:**

In this paper, the authors introduce a simple mixture-of-experts model, by greatly simplifying the routing mechanism: experts are randomly activated both at train and inference time. A consistency loss function is added for training the proposed models, enforcing all experts to make consistent predictions. The proposed method, called THOR, is evaluated on machine translation tasks, including multi-lingual MT, and outperforms the recently proposed Switch Transformer MoE.

The reviews note that the paper is well written and easy to follow, and that the proposed method is simple. While the results look promising, the reviewers also raised concerns regarding comparisons to previous work, some of which were addressed in the rebuttal. Finally, a reviewer raised the concern that this method is related to ensembles, which work well for machine translation, but are not discussed or compared to. For these reasons, I believe that the paper is borderline, leaning toward acceptance.